# Modelling and Validation of a Grid-Connected DFIG by Exploiting the Frequency-Domain Harmonic Analysis

**Emmanuel Hernández-Mayoral** [1,*], **Reynaldo Iracheta-Cortez** [2] , **Vincent Lecheppe** [3]
**and Oscar Alfredo Jaramillo Salgado** [4]

[1] CONACyT Attached to Instituto de Energías Renovables, Universidad Nacional Autónoma de Mexico, 62580 Temixco, Mor., Mexico

[2] CONACyT Attached to Isthmus University, Av. University, Azteca, 70760 Tehuantepec, Oax., Mexico; reynaldo.iracheta@cimat.mx

[3] Université de Lyon, INSA Lyon, ECL, Université Claude Bernand Lyon 1, CNRS UMR5005, Ampère, 69100 Villeurbanne, France; vincent.lechappe@insa-lyon.fr

[4] Instituto de Energías Renovables, Universidad Nacional Autónoma de Mexico, 62580 Temixco, Mor., Mexico; ojs@ier.unam.mx

* Correspondence: emhema@ier.unam.mx; Tel.: +45-777-362-0090 (ext. 38021)

**Abstract:** Wind Energy Conversion Systems (WECS) based on a Doubly-Fed Induction Generator (DFIG) represent the most common configuration employed in wind turbines. These systems involve injecting harmonic currents toward an electrical grid from a back-to-back power converter, potentially creating voltage distortions. To assess this phenomenon, a case study of a 3 kW DFIG-based wind turbine connected to the electrical grid is presented for analysis in the harmonic domain. Initially, a DFIG-based load flow analysis for determining the operating conditions is tackled at the fundamental frequency. Then, the modelling of a DFIG under steady-state operating conditions at harmonic frequencies is analyzed discussing its characteristics in the harmonic domain. The high-frequency harmonics in the output voltage of a pulse width modulation-driven inverter feeding the rotor windings of a DFIG and its connection to a three-winding transformer are also analyzed. This investigation produced a complete model of the DFIG connected to the electrical grid. The results demonstrated that although a considerable harmonic contribution up to the 25th order exists, it remains harmless since it is below 5%, according to the Std. IEEE 519.

**Keywords:** Doubly-Fed Induction Generator (DFIG); back-to-back power converter; harmonic analysis; steady-state studies; load flow calculation

## 1. Introduction

Wind energy generation is developing at a faster rate than the rest of the other renewable energy sources, of which Germany, Spain, Denmark, and the United States are the world's major producers [1]. Conversely, Latin America and the Caribbean are currently the world regions with the lowest wind energy growth and installed capacity. Cuba, however, with its moderate installed capacity is pursuing a program for the development of wind energy. Similarly, Mexican projects are improving their installed capacity by 5500 MW through with 52 new renewable energy generation plants by 2023 [2]. The development of wind energy generation worldwide is also accompanied by advances in wind turbine technology because the knowledge of the technology type associated with wind turbines was utilized in a wind park to determine power quality.

In general, a wind turbine connected to an electrical grid involves several problems associated with the electromechanical energy conversion element, the random characteristics of wind, and the grid conditions. The impact of these problems varies depending on the characteristics of the electrical grid in connection with the wind turbine. In Mexico, a major issue that arises for such a type of connection is harmonic distortion caused by the utilization of the variable-velocity wind turbines, in which power converters for coupling to the electrical grid generate harmonics that can raise concerns in an electrical system involving harmonic resonance.

A type 3-DFIG essentially consists of a wound rotor induction machine with slip rings and a stator, which is directly connected to the electrical grid, whereas the rotor is interfaced to the system by a partially rated back-to-back (B2B) power converter, which handles a fraction (25–30%) of the total DFIG power to achieve full control of the generator [3]. Because this amount of power is significant, the effects of the harmonic distortion introduced by the B2B power converter toward the electrical grid are critical, so that special concerns are focused on power electronics. The DFIG interconnected to an electrical grid is commonly studied for load flow analysis [4–9], disturbances in the electrical grid [10–15], stability [16–20], and voltage variation [21–24], with very few studies on harmonic propagation. For example, a second-order model for the prediction of the response of a DFIG wind turbine to grid disturbances is simulated and verified experimentally. Basically, steady-state impact, such as flicker emission, reactive power, and harmonic emission, are measured and analyzed. The results of this whole operating range, and the current THD is always lower than 5% [25]. Other research investigations were compared with the response to a fixed-speed wind turbine. It is found that the flicker emission is very low, the reactive power is close to zero and the subject exposes a converter connected to the electrical grid for the harmonic Linear Time varying Periodically (LTP) system. In this investigation, an electrical grid validates the method [26]. Another article mentions that the harmonic transient analysis of the analysis of a Harmonic State Space (HSS) small-signal model, which is modeled from connected to converter harmonic matrix investigated to analyze the harmonic time-domain simulation results, are represented through the electrical grid of HSS maintenance of the basic parameters of the voltage and frequency, and requires very demanding levels of power quality to ensure the operation of the electrical power behavior interaction and the dynamic transfer procedure. Frequency-domain as well as modeling is used to verify the theoretical analysis. Experimental results are also included, since phenomena such as the continuity of the service, grid stability, and the wind turbine generators requires reliable models for reproducing harmonic dynamics accurately and for ensuring that such models perform close to the actual components. For this reason, in this research a flexible extended harmonic domain (FEHD) technique together with the Time-Domain (TD) are interfaced to produce a hybrid TD/FEHD, which is used to model a synchronous-based wind turbine generator. The obtained results are validated with PSCAD EMTDC® and are verified with experimental data, revealing a very good agreement [27]. Finally, a study was published regarding the power converters used for coupling wind turbines to the electrical grid.

This study analyzed the generation of high frequency harmonics under the influence of controllers. In this addition, this article studies the relationship between the output and the harmonic source at grid-side and rotor-side converters based on their control features in the DFIG system. The effectiveness of the proposed model is verified through the 2 MW DFIG real-time hardware-in-the-loop test platform by StarSim® software and real test data, respectively [28].

After analyzing the specialized literature, it is concluded that work has been carried out for the harmonic analysis of wind turbines connected to the electrical grid. However, they do not consider the impact of high-order inter-harmonics on the system. So, the research in this article presents a clear model analysis of frequencies generated by the DFIG for the harmonic and non-harmonic analysis of the DFIG connected to the electrical grid. This investigation is powered by exploiting the frequency-domain harmonic analysis and by considering the impact of the high frequency components on the power quality system. Similarly, wind power plants, like all other sources of electricity generation, should contribute to maintaining the stable grid under any operating condition.

The major contributions of the paper are pointed out below:

1.  This analysis provides a complete model of the DFIG steady-state operation that is directly applicable to the load flow analysis. It allows a complete knowledge of the steady-state operation condition of a DFIG considering the wind speed as input value.
2.  Considering the attained results from the load flow analysis, a model of the DFIG steady-state operation in the harmonic domain is completed; this model adopts the sum of both effects in each voltage source, that is, both the stator and the rotor, for obtaining a complete solution for a DFIG.
3.  By setting up an experimental test, the obtained results for the proposed model in the steady-state and transient-state of the DFIG under non-sinusoidal voltage conditions are validated with those resulting from the hardware implementation, demonstrating the effectiveness of the model presented.
4.  Furthermore, this analysis provides a clear analysis of the frequencies generated by the DFIG in the steady-state resulting in a complete and proper model for harmonic and non-harmonic analysis of the DFIG connected to the electrical grid.

This paper is structured as follows. In Section 2, the model of a DFIG and its analysis at a fundamental frequency is described. A DFIG-based load flow analysis is addressed in Section 3. In Section 4, the DFIG complete solution for harmonic propagation is derived. A topology of the B2B power converter for grid connection purposes is described in Section 5. In Section 6, the results obtained with the proposed model made in MATLAB-Simulink® are validated by matching those obtained from real lab measurements and with a complete induction machine model for transient studies. Finally, concluding remarks are given in Section 7.

## 2. Analysis of the DFIG at Fundamental Frequency

The general configuration of a DFIG-based wind turbine connected to an electrical grid is shown in Figure 1. The electrical single-equivalent phase circuit of the DFIG is illustrated in Figure 2.

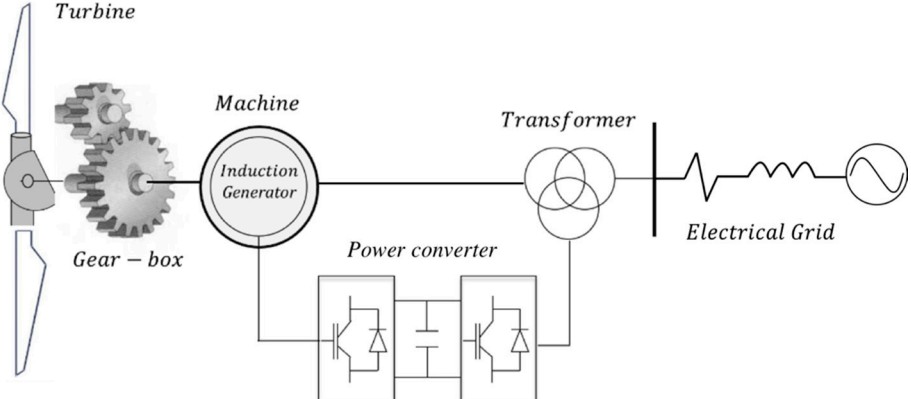

**Figure 1.** Configuration of a DFIG connected to the electrical grid.

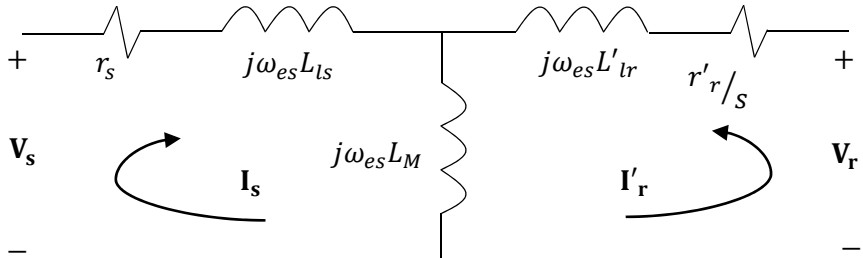

**Figure 2.** Equivalent circuit of the induction generator in steady-state seen from stator.

The circuit displays a conventional steady-state equivalent model, with all parameters referring to the stator. The parameters of Figure 2 were previously described. The equation describing the system is given by Equation (1),

$$\begin{aligned} \mathbf{V_s} &= -(r_s + j\omega_{es}L_{ls})\mathbf{I_s} + j\omega_{es}L_M\mathbf{I}'_{\mathbf{r}} \\ \mathbf{V}'_{\mathbf{r}} &= r'_r/s + j\omega_{es}(L'_{lr} + L_M)\mathbf{I}'_{\mathbf{r}} - j\omega_{es}L_M\mathbf{I_s} \end{aligned}$$ 

(1)

where $s$ is defined as the relative velocity between the magnetic field produced by the currents injected in the stator and the mechanical velocity of the rotor using Equation (2),

$$s = \frac{\omega_{es} - \omega_r}{\omega_{es}}$$ 

(2)

From Equation (1), the stator and rotor current phasors in the time-domain are obtained by Equation (3):

$$\begin{aligned} \mathbf{i}_s &= |I_s|\cos(\omega_{es}t + \varphi_s) \\ \mathbf{i}_r &= |I'_r|\cos\left(s\omega_{es}t + \varphi'_r - \theta_{eff}\right) \end{aligned}$$ 

(3)

The importance of the slip is analyzed as follows: if the machine is stopped, the shaft's mechanical velocity is zero ($\omega_r = 0$), resulting in a slip value of 1 ($s = 1$) obtained from Equation (2). In this condition the machine operates as a blocked rotor. On the contrary, if the rotor rotates at the magnetic field velocity, then the slip is zero ($s = 0$). In this case, the magnetic field velocity is termed the synchronous velocity of the machine. If the rotor rotates higher than the synchronous velocity, the slip becomes negative, and the machine operates as a generator. Conversely, if the velocity of the magnetic field of the stator exceeds that of the rotor, the machine operates as a motor. Therefore, the slip defines the operation of the machine. Therefore, when all parameters of the induction generator and the electrical grid are known, the slip determines the point of operation.

To ensure a complete analysis of the DFIG, the effects of the generator's rotor must be considered. The steady-state equivalent circuit model is depicted in Figure 3, with all parameters referring to the rotor [3].

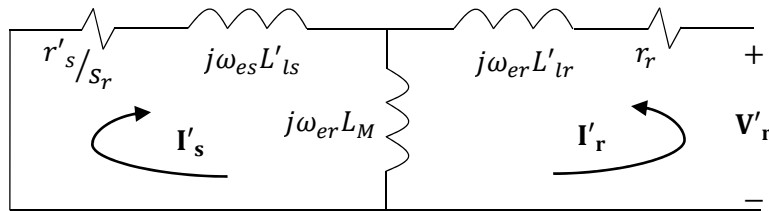

**Figure 3.** Steady-state equivalent circuit of the generator seen from the rotor.

Equation (4) describes the system given by,

$$\begin{aligned} 0 &= r'_s/s_r + j\omega_{er}(L'_{ls} + L_M)\mathbf{I}'_s - j\omega_{er}L_M\mathbf{I}_r \\ \mathbf{V_r} &= r_r + j\omega_{er}(L_{lr} + L_M)\mathbf{I}_r - j\omega_{es}L_M\mathbf{I}'_s \end{aligned}$$ 

(4)

Since the rotor is moving, the slip seen from it is given by Equation (5),

$$s_r = \frac{\omega_{er} + \omega_r}{\omega_{er}}$$ 

(5)

Considering Equation (4), the stator and rotor currents phasors in the time domain are obtained by Equation (6),

$$\begin{aligned} \mathbf{i}_r &= |I_r|\cos(\omega_{er}t + \varphi_r) \\ \mathbf{i}_s &= |I'_s|\cos\left(s_r\omega_{er}t + \varphi'_s + \theta_{eff}\right) \end{aligned}$$ 

(6)

Thus, the complete solution for a DFIG at fundamental frequency considering the stator and rotor effects is given by Equation (7),

$$\begin{aligned}
\mathbf{i_s} &= |I_s|\cos(\omega_{es}t + \varphi_s) + |I'_s|\cos\left(s_r\omega_{er}t + \varphi'_s + \theta_{eff}\right)\\
\mathbf{i_r} &= |I_r|\cos(\omega_{er}t + \varphi_r) + |I'_r|\cos\left(s\omega_{es}t + \varphi'_r - \theta_{eff}\right)
\end{aligned} \tag{7}$$

## 3. DFIG-Based Load Flow Analysis

A commonly applied approach for determining operating conditions in power systems is the load flow analysis. From the conventional topology and the electrical grid parameters, load flow analysis establishes unknown electrical variables like bus voltages (magnitude and angle), powers through the links, and active and reactive powers. Traditionally, an induction machine was an additional load within the load flow treated as a constant PQ load. The alternative energy sources revived the use of the induction machine as a generator, however, with significant interest in evaluating the performance of such a unit. Thereby, the most refined models are incorporated in the load flow analysis, considering a more realistic mode of the induction machine's characteristics.

For load flow analysis, the DFIG is modeled as a PV or PQ node [29–31], depending on the control scheme envisaged. A PV representation is applied when the control objective focuses on terminal voltage control. This is advantageous because only the wind velocity is needed as an input variable. In contrast, a PQ representation is employed when the control objective focuses on controlling the Power Factor (PF). Through load flow analysis, the node voltages of the DFIG's connections, the active power delivered to the grid, and the reactive power consumed by the DFIG are obtained. These results verify whether the DFIG is operating at its nominal power. If it is operating below its nominal power, the rotor velocity is calculated using the power curve generated against the rotor velocity provided by the manufacturer. According to the generated power, the electric torque is determined by $T_e = P_e/\omega_r$. Bearing the rotor velocity, the slip is calculated using Equation (1). Finally, the variables resulting from the initialization method are calculated. If the DFIG operates at its nominal velocity, however, the nominal rotor velocity, the wind velocity, or the inclination angle of the blade are required.

The electric torque, the slip and the resulting variables are calculated. The mechanical power $P_m$ provides the power balance in the DFIG and the electrical values in the stator $P_s$, rotor $P_r$, and losses $P_{loss}$, are given by Equation (8).

$$P_m = P_s + P_r + P_{loss} = P_s + P_r + \left(I_s{}^2 r_s + I_r{}^2 r_r\right) \tag{8}$$

Additionally, the stator and rotor powers are given by Equation (9),

$$\begin{aligned}
P_s &= V_s I_s \cos(\delta_s - \varphi_s)\\
P_r &= V'_r I'_r \cos(\delta_r - \varphi_r)
\end{aligned} \tag{9}$$

Thereby, the total active power injected toward the electrical grid by the DFIG is given by Equation (10),

$$P = P_s + P_r \tag{10}$$

The reactive power $Q$ is controllable at specified ranges of the agreed exchange between the generator and the electrical grid. That is, if $Q$ is predetermined at zero, then the power exchange occurs through the stator of the generator, producing a reactive power given by Equation (11).

$$Q = V_s I_s \sin(\delta_s - \varphi_s) \tag{11}$$

A relationship is considered between the mechanical power and the rotor velocity of the generator, which is established in Equation (12), as seen in Figure 4.

$$P_m = \gamma_0(1-s)^3 \tag{12}$$

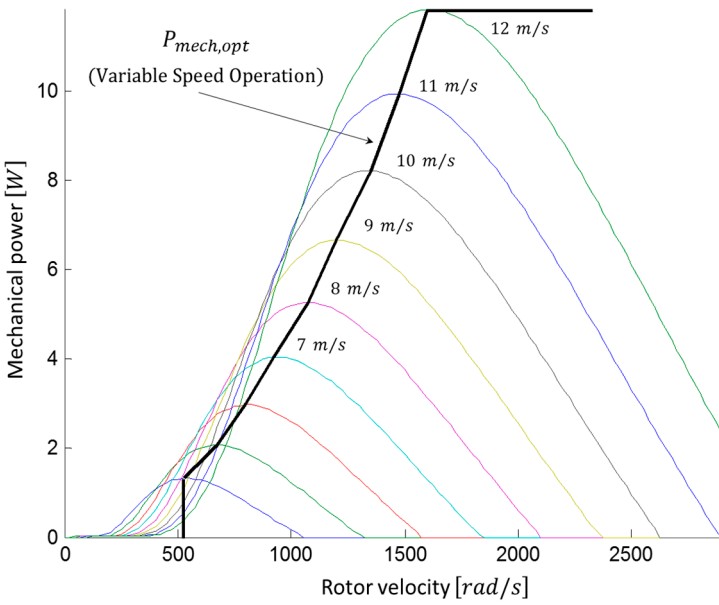

**Figure 4.** Mechanical power of the DFIG as a rotor speed function.

Regarding Equation (12), the mechanical power $P_m$ [W] extracted from the wind can be calculated by Equation (13)

$$P_m = \frac{1}{2}\rho A V_w^3 C_p \tag{13}$$

where $\rho$ is the air density $\left[\text{kg/m}^3\right]$, $A$ the area covered by the rotor $\left[\text{m}^2\right]$, $C_p$ is the power coefficient and $V_w$ is the wind speed upstream the rotor [m/s]. The power coefficient $C_p$ depends on the values of the pitch angle $\beta$ [deg] and the tip speed ratio $\lambda$, defined in Equation (14):

$$\lambda = \frac{v_t}{V_w} \tag{14}$$

where $v_t$ is the blade tip speed [m/s]. The three modes of operating the speed control of a DFIG include:

- Below the synchronous speed (for low wind speeds). For this case, $C_{p,opt}$ cannot be reached and the DFIG is operated at a constant speed ($\omega_{r,\,min}$) and $\beta = 0$;
- At maximum power production;
- Above the synchronous speed (for high wind speed). In this case, the speed control optimizes the power extraction from the wind between the minimum and maximum wind speeds given by Equation (15), using an optimal value for the power coefficient $C_p = C_{p,opt}$ and $\beta = 0$,

$$P_m = \frac{1}{2}\rho A V_w^3 C_{p,\,opt} \tag{15}$$

where $\gamma = (1/2)\rho A C_{p,opt}$. In addition, if it is multiplied by the cube of the wind speed, we obtain, $P_m = \gamma V_w^3$. Now, by substituting the wind speed variable produced in Equation (14), it is obtained that

$$P_m = \gamma\left(\frac{v_t}{\lambda}\right)^3 = \gamma'\omega_r^3 = \gamma_0(1-s)^3 \tag{16}$$

Considering the Equations (1) to (12), a set of equations describing the DFIG in a steady-state is obtained. To solve these, another equation set is derived, taking the real and imaginary parts in Equations (1) and (3), obtaining the following:

$$F_1 = V_s \cos \delta_s - r_s I_s \cos \varphi_s + \omega_{es}(L_{ls} + L_M)I_s \sin \varphi_s + \omega_{es}L_M I'_r \sin \varphi_r \tag{17}$$

$$F_2 = V_s \sin \delta_s - r_s I_s \sin \varphi_s - \omega_{es}(L_{ls} + L_M)I_s \cos \varphi_s - \omega_{es}L_M I'_r \cos \varphi_r \tag{18}$$

Similarly, the Equations (19) and (20) are obtained:

$$F_3 = V'_r \cos \delta_r - r'_r I'_r \cos \varphi_r + s\omega_{es}(L'_{lr} + L_M)I'_r \sin \varphi_r + s\omega_{es}L_M I_s \sin \varphi_s \tag{19}$$

$$F_4 = V'_r \sin \delta_r - r'_r I'_r \sin \varphi_r - s\omega_{es}(L'_{lr} + L_M)I'_r \cos \varphi_r - s\omega_{es}L_M I_s \cos \varphi_s \tag{20}$$

From the Equations (9) and (10), the function F5 is

$$F_5 = P - V_s I_s \cos(\delta_s - \varphi_s) - V_r I_r \cos(\delta_r - \varphi_r) \tag{21}$$

Thus, from Equation (11), the function F6 is,

$$F_6 = Q - P_s = Q - V_s I_s \cos(\delta_s - \varphi_s) \tag{22}$$

Finally, from the Equations (8), (10), and (12) we obtain

$$F_7 = P - \gamma_0(1 - s)^3 + \left(I_s^2 r_s + I_r^2 r_r\right) \tag{23}$$

In this manner, eleven variables in the previous set of seven nonlinear equations are as following: $U_s$, $\delta_s$, $U_r$, $\delta_r$, $I_s$, $\varphi_s$, $I_r$, $\varphi_r$, $s$, $P$, and $Q$. To solve the Equations (17)–(23), four variables with known value are required. By selecting $V$, $\delta$, $P$, and $Q$ as fixed variables, *ss* solutions for the Equations (17)–(23) are obtained using the Decoupled Newton–Raphson method expressed as

$$\Delta x = J^{-1} \Delta F \tag{24}$$

where $\Delta x$ and $\Delta F$ are the incremental variables defined by the Equations (25) and (26):

$$\Delta F = \Delta(F_1, F_2, F_3 \ldots F_7)^T \tag{25}$$

$$\Delta x = \Delta(V_r, \delta_r, I_r, \varphi_r, I_s, \varphi_s, s)^T \tag{26}$$

The Jacobian matrix $J$ in Equation (27) contains partial derivatives of the functions in the Equations (17)–(23) for the variables of $\Delta x$ in Equation (26). Using adequate initial conditions, the solution of (26) is estimated using a few numbers of iterations. The calculation of the initial conditions depends on the results of the load flow analysis. Thus, the model is calculated from the output of the machine to the inputs. The following assumptions are established:

$$J = \begin{pmatrix} \dfrac{\partial F_1}{\partial U_r} \dfrac{\partial F_1}{\partial \delta_r} & \cdots & \dfrac{\partial F_1}{\partial s} \\ \dfrac{\partial F_2}{\partial U_r} \dfrac{\partial F_2}{\partial \delta_r} & & \\ \vdots & \ddots & \vdots \\ \dfrac{\partial F_7}{\partial U_r} \dfrac{\partial F_7}{\partial \delta_r} & \cdots & \dfrac{\partial F_7}{\partial s} \end{pmatrix} \tag{27}$$

- The system is modeled in steady-state with all derivatives equal to zero.
- Magnetic saturation is neglected.
- The flow distribution is sinusoidal.
- Finally, the losses in the B2B power converter are ignored.

### 3.1. PQ Bus Model

The reactive power requirements of the system depend on the wind turbine involved. Wind turbines (WTs) based on the DFIG, for instance, require reactive compensation. The control scheme design defines the relationship between the following variables: active and reactive power, power delivered, and nodal voltage. The capacitor banks or reactive power sources are individually added to wind turbines for supplying the reactive power deficit locally without importing from other parts of the electrical grid. The fluctuations of the active and reactive power caused by the wind speed are necessary since the active power depends on the speed [9]. Therefore, the three variables required as input for the PQ bus model are: wind speed $V_W$, active $P$ and reactive Q power. Thus, the Equations (17)–(23) are solved, with the unknown values given in Equation (26). This process provides the machine's operating point in a steady-state.

### 3.2. PV Bus Model

In the PV bus model, the active power of the generator $P$, the stator voltage $V_s$, and the wind speed $V_W$ are input variables. The active power of the generator $P$ remains constant in the load flow analysis for calculating the reactive power and the slip. The difference between the reactive power and that obtained in the load flow analysis is modeled as an admittance in the admittance characteristic of the load node. The difference between the generator's reactive power and the initial reactive power of the node is obtained by adding a reactive power compensation in parallel. As in the previous method, the set of the Equations (17)–(23) is solved, with the unknown values given in Equation (26). The result of this process also provides the machine's operating point in steady-state. The characteristics of the PQ and PV methods for the load flow analysis with the input and the unknown variables derived from the solution process are presented in Table 1.

**Table 1.** PQ and PV methods for Wind Turbines based on DFIG.

| Bus Type | Specified Variable | | | | Variables Obtained in the Solution Process | | | | | | | | | |
|----------|-------|---|---|-------|---|-------|------------|-------|------------|-------|-------------|-------|-------------|---|
| | $V_W$ | $P$ | $Q$ | $V_s$ | $Q$ | $V_s$ | $\delta_s$ | $V_r$ | $\delta_r$ | $I_s$ | $\varphi_s$ | $I_r$ | $\varphi_r$ | $s$ |
| PQ bus | • | • | • | | | • | • | • | • | • | • | • | • | • |
| PV bus | • | • | | • | • | | • | • | • | • | • | • | • | • |

### 3.3. Wind Speed Analysis

For conventional wind turbines, the operating point can be obtained by an iterative process combined with the load flow analysis considering all characteristics provided by the manufacturer. To obtain all electrical variables of the operating point in steady-state for the DFIG, however, the mechanical power based on the wind speed given by the manufacturer should be estimated, for controlling the reactive power. This procedure presents the active power $P$ and the stator voltage ($\mathbf{V_s} = V_s \angle \delta_s$) as unknown variables in the model. By establishing a new equation for F7, given by: $P - P_m + \left( I_s^2 r_s + I_r^2 r_r \right)$, the value of $P$ is known, but $P$ is a proposed value. This could be resolved if the process is iterative and a more accurate calculation is performed. Another way to conduct the simulation is by assuming the slip value, which depends on the control mode since the slip value is different for each wind speed. Considering the above, the slip value is known once the input value of the wind speed is available. This facilitates the calculation since the unknown in Equation (26) is

eliminated and Equation (27) is a $6 \times 6$ matrix. Since all DFIG variables are known, the harmonic analysis is performed as described in subsequent sections.

## 4. Harmonic Analysis of the DFIG

DFIG generates harmonics through the stator or rotor voltage sources: the harmonic frequencies for stator and rotor are $f_{sh} = hf_{es}$ and $f_{rh} = hf_{er}$, respectively. Two fundamental frequencies are available: one in the stator ($f_{es}$) and another in the rotor ($f_{er}$). To perform the frequency-domain analysis of the DFIG at harmonic frequencies by analyzing the generator with its independent voltage sources, it is necessary that only the effects of the stator are considered. This is done by feeding the DFIG by the stator with frequency $f_{sh}$ and keeping the rotor short-circuited (Figure 5). Equation (28) describes the circuit as follows:

$$
\begin{aligned}
U_{sh} &= -(r_s + jh\omega_{es}L_{ls})I_{sh} + jh\omega_{es}L_M I'_{rh} \\
0 &= r'_r/s_h + jh\omega_{es}(L'_{lr} + L_M)I'_{rh} - jh\omega_{es}L_M I_{sh}
\end{aligned}
\tag{28}
$$

where the harmonic slip seen from stator is given by Equation (29)

$$
s_h = \frac{\pm h\omega_{es} - \omega_r}{\pm h\omega_{es}}
\tag{29}
$$

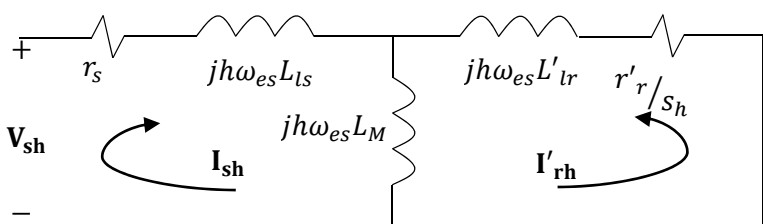

**Figure 5.** Harmonic equivalent circuit of the induction generator in steady-state seen from stator.

In Equation (29), the (+) and (−) represent positive and negative harmonic sequences. By solving the Equation (28), the stator and rotor harmonic current phasors in the time-domain are obtained by Equation (30)

$$
\begin{aligned}
i_{sh} &= |I_{sh}|\cos(h\omega_{es}t + \varphi_{sh}) \\
i_{rh} &= |I'_{rh}|\cos\left(s_h h\omega_{es}t + \varphi'_{rh} \pm \theta_{eff}\right)
\end{aligned}
\tag{30}
$$

where $\pm \theta_{eff}$ is (−) for the positive and (+) for negative sequence. On the contrary, feeding the DFIG by the rotor with frequency $f_{rh}$ keeps the stator in short-circuit (Figure 5), and the equation (31) represents the circuit,

$$
\begin{aligned}
0 &= r'_s/s_{rh} + jh\omega_{er}(L'_{ls} + L_M)I'_{sh} - jh\omega_{er}L_M I_{rh} \\
V_{rh} &= r_r + jh\omega_{er}(L_{lr} + L_M)I_{rh} - jh\omega_{es}L_M I'_{sh}
\end{aligned}
\tag{31}
$$

where the harmonic slip seen from rotor is given by Equation (32)

$$
s_{rh} = \frac{\pm h\omega_{er} + \omega_r}{\pm h\omega_{er}}
\tag{32}
$$

Similarly, the (+) and (−) signs in Equation (32) represent the positive and negative harmonic sequence, respectively. By considering Equation (4), the stator and rotor current phasors in the time-domain are obtained by Equation (33),

$$
\begin{aligned}
i_{rh} &= |I_{rh}|\cos(h\omega_{er}t + \varphi_{rh}) \\
i_{sh} &= |I'_{sh}|\cos\left(s_{rh}h\omega_{er}t + \varphi'_{sh} + \theta_{eff}\right)
\end{aligned}
\tag{33}
$$

where $\pm \theta_{eff}$ is (−) for the positive and (+) for the negative sequence. Finally, the effects of the harmonic voltage sources in the stator and rotor summed to provide the complete solution for the DFIG are given by Equations (34) and (35):

$$\mathbf{i_{sh}} = |I_{sh}|\cos(h\omega_{es}t + \varphi_{sh}) + |I'_{sh}|\cos(s_{rh}h\omega_{er}t + \varphi'_{sh} \pm \theta_{eff}) \tag{34}$$

$$\mathbf{i_{rh}} = |I_{rh}|\cos(h\omega_{er}t + \varphi_{rh}) + |I'_{rh}|\cos(s_{h}h\omega_{es}t + \varphi'_{rh} \pm \theta_{eff}) \tag{35}$$

Regarding the zero sequence components, in a DFIG only the wye connections with insulated neutral and delta connect the stator phases of the induction generator. The wye connection with the neutral connected to the electrical grid is not used in these machines. This indicates that homopolar currents are absent in the stator of these machines. Therefore, Figure 2 is invalid since the induction generator operates as two decoupled windings for the zero sequence. Equation (36) represents the components of the zero sequence in the induction generator as follows:

$$\begin{aligned} V_{sh} &= (r_s + jh\omega_{es}L_{ls})I_{sh} \\ V_{rh} &= (r_s + jh\omega_{es}L_{ls})I_{rh} \end{aligned} \tag{36}$$

By considering Equation (36), the stator and rotor harmonic current phasors in the time-domain are obtained by Equation (37)

$$\begin{aligned} \mathbf{i_{sh}} &= |I_{sh}|\cos(h\omega_{es}t + \varphi_{sh}) \\ \mathbf{i_{rh}} &= |I_{rh}|\cos(h\omega_{er}t + \varphi_{rh}) \end{aligned} \tag{37}$$

In summary, if voltage sources are considered in balanced conditions at the fundamental frequency, as well as the harmonic frequencies in each winding, then the general solution is given by Equation (38).

$$
\begin{aligned}
\mathbf{i_s} = &\sum_{h=1}^{H} |I_{sh}|cos(h\omega_{es}t + \varphi_{sh}) + \sum_{h=1,seq(+)}^{H} |I'_{sh}|cos(s_{rh}h\omega_{es}t + \varphi'_{sh} + \theta_{eff}) \\
&+ \sum_{h=1,seq(-)}^{H} |I'_{sh}|cos(s_{rh}h\omega_{es}t + \varphi'_{sh} - \theta_{eff})
\end{aligned}
\tag{38}
$$

Equation (38) is obtained from Equations (3), (6), (30), (33), and (37) and is interpreted as follows: first, all current harmonics are generated by a non-sinusoidal voltage source in the stator for positive, negative, and zero sequences. All current harmonics are then induced on the rotor from the stator of the positive sequence. Finally, all current harmonics are induced on the rotor from the stator of negative sequence. The same interpretation is used for Equation (39).

$$
\begin{aligned}
\mathbf{i_r} = &\sum_{h=1}^{H} |I_{rh}|\cos(h\omega_{er}t + \varphi_{rh}) + \sum_{h=1,seq(+)}^{H} |I'_{sh}|\cos(s_{h}h\omega_{es}t + \varphi'_{rh} - \theta_{eff}) \\
&+ \sum_{h=1,seq(-)}^{H} |I'_{rh}|\cos(s_{h}h\omega_{es}t + \varphi'_{rh} + \theta_{eff})
\end{aligned}
\tag{39}
$$

## 5. DFIG Connected at Electrical Grid

The integration of wind farms to the electrical grid is a complicated problem. Numerous studies related to the impact of wind penetration on power systems have been conducted, including those related to stability [32–35], and electrical protection response and harmonic effects in the grid [36–39]. This article, however, focuses on the analysis of the harmonics generated by the B2B power converter and its interconnection to the electrical grid.

### 5.1. B2B Power Converter Analysis

The B2B power converter topology used in this analysis comprises a rectifier and an inverter connected in series to a DC bus in parallel with a capacitor. This converter type is bidirectional,

allowing operation in the four quadrants of the XY plane, and, therefore, both converters operate as rectifiers or inverters. The difference between the rectifier and the inverter is the definition of the power sign. The active and reactive powers through the rotor and the stator are controlled by adjusting the amplitude, phase and frequency of the voltage introduced to the rotor. The rotor-side converter provides a three-phase voltage at a variable amplitude and frequency, controlling the generator torque and the reactive power exchange between the stator and the electrical grid. The grid-side converter exchanges the active power with the electrical grid obtained or injected by the rotor-side converter from the rotor. The output frequency of the grid-side converter is constant, while the output voltage varies depending on the exchange of the active and reactive powers with the electrical grid. Because of the greater flow of power injected by the stator generator, the B2B power converter is designed to operate between 25% and 30% of the nominal power of the generator. This reduces the design cost of the power converter and the losses of the generation scheme. Thus, the wind turbine's generated power is distributed between the stator and the supply through the rotor. The three-phase B2B power converter is shown in Figure 6.

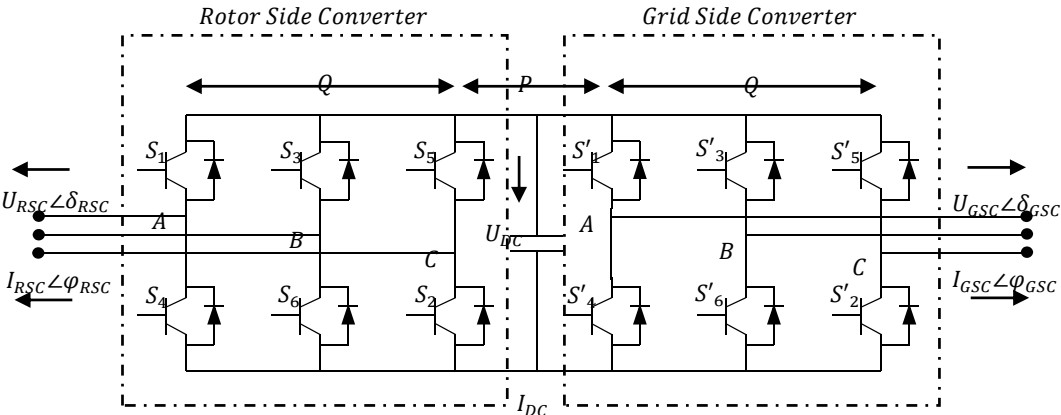

**Figure 6.** B2B power converter configuration.

The converter is modeled with bidirectional switches, where the ideal switch is created by a semiconductor with an anti-parallel diode allowing current to flow in both directions. The semiconductor used is an isolated gate bipolar transistor (IGBT) with ideal operation. For the synchronization and power injection of the B2B power converter toward the electrical grid, it is necessary to know the phase's sequence and to synchronize one phase of the grid by the phase-locked loop (PLL) method. This method requires estimating the grid angle to accomplish the necessary transformations using the d-q reference frame of the grid voltage.

### 5.1.1. Loop Controller

There are many papers in the literature on control strategies for controlled current three-phase power converters. Basically, the control problem is identical to synchronous-machine control. Since the B2B power converter control method is not the main point of this work, only a brief description of the control strategy implemented in this analysis will be mentioned. In this paper, the synchronous PI controllers are used. This is one of the most popular control methods. The idea is to transform currents and voltages into a rotating reference frame, where the controlled currents are constant in steady-state and use ordinary PI controller outputs back to the fixed reference frame. The drawbacks are the nonlinearities introduced in the transformation and the less than optimal control performance. The advantage is that it is simple to understand and simple to implement. Synchronous PI controllers work with averaged voltages and currents. The converter is controlled via a PWM modulator that generates a switching pattern that corresponds to the desired average voltage. In this paper, the line current controller is assumed to be a synchronous PI controller. Its properties are well known in the power electronics area. Furthermore, by assuming this type of controller, the three-phase converter

can be modeled as two separate four quadrant converters: one for the active power and one for the reactive power. The reactive power is assumed to be zero, which means that this part can be removed. It is important to consider that the controlled system is nonlinear, but the large-signal stability is not examined. The stability of the closed-loop system should be examined with methods for non-linear systems. The load power is a factor in all coefficients of the characteristics equation of the closed-loop system, and the poles of the linearized system move around with the load of the B2B power converter. Consequently, the controller parameters must be chosen to keep all poles in the LHP of the complex plane under all load conditions.

### 5.1.2. Modulation Method

The B2B power converters have as their main component a voltage source converter, which uses pulse width modulation (PWM) for controlling purposes. This technique uses power semiconductors that operate at high frequencies. In this paper, the sinusoidal PWM (SPWM) is used since it is mature technology. In this method a triangular wave is compared with a sinusoidal wave. For the SPWM technique, with an amplitude modulation index less than one, the fundamental voltage amplitude varies linearly with this modulation index, but the fundamental frequency is less. When you have an amplitude modulation index greater than one, the amplitude also increases, leading to overmodu-lation, and the output waveform contains many harmonics. The dominant harmonics in the linear range may not be so in overmodulation; the amplitude of the fundamental component does not vary linearly with the amplitude modulation index. In power applications, the overmodulation region should be avoided as much as possible to minimize distortion in the output voltage. Within the MATLAB-Simulink® software, the SPWM technique is developed using some blocks that have already been designed in the master library. The carrier signal operates at 5 kHz and is compared with the modulated signals at 60 Hz.

### 5.2. Electrical Grid Configuration

In this analysis, the electrical grid is represented by a Thevenin equivalent circuit, consisting of a constant magnitude short-circuit power $S_{cc} = 10$ kVA and a grid impedance angle $\vartheta_{cc} = 80°$. For simplicity, the electrical system is represented by a straightforward model using the commercial program MATLAB-Simulink®. This representation is convenient in the absence of a specific grid, with generalized conclusions being sought. The interconnection diagram of the DFIG with the electrical grid is displayed in Figure 7. The power exchange of the machine is carried out by the stator and the B2B power converter on the grid side converter. Meanwhile, the total active power of the machine is equal to the sum of the stator power and B2B power converter power on the grid-side converter. The grid-side converter is assumed to operate with a unit power factor so that the grid side converter only exchanges active power with the electrical grid, and the reactive power exchange is with the stator circuit. To connect the DFIG with the electrical grid, a transformer is used to adjust the values of the generated voltage to the distribution value of the electrical grid. This establishes an interconnection point where the stator voltage and the converter voltage of the grid-side converter are in low voltage and have different magnitudes.

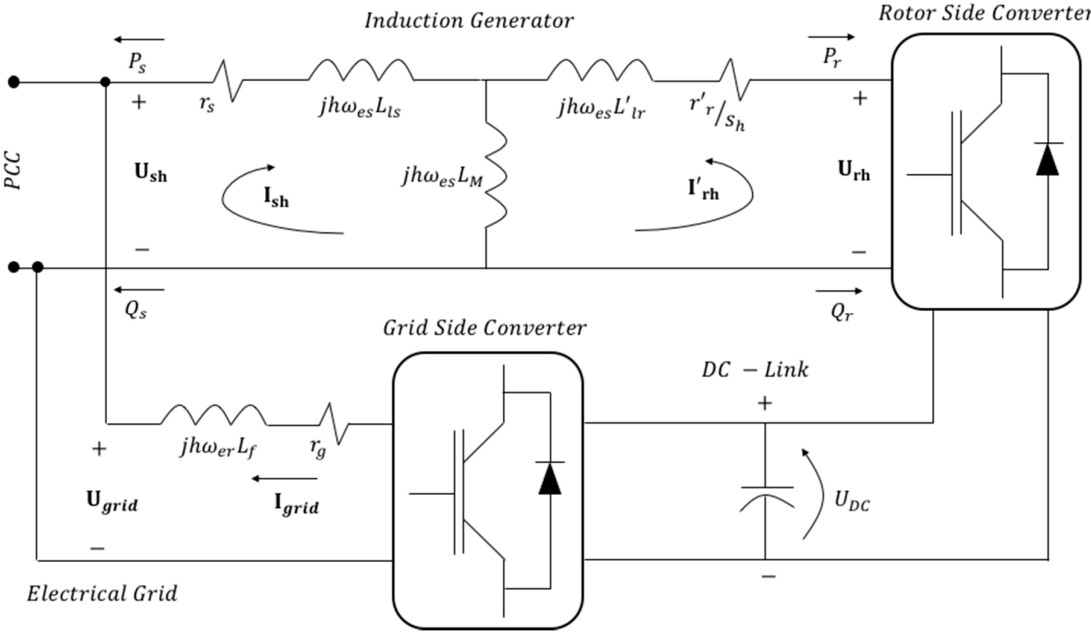

**Figure 7.** Interconnection diagram of the DFIG with the electrical grid.

## 6. Model Validation

An experimental test is performed on a three-phase DFIG of 3 kW, 230/400 V, and 11.5 A to validate the proposed model. The PWM converters used were standard 4.5 kW with a maximum switching frequency of 5 kHz. The schematic and block diagram of the experimental rig are exhibited in Figures 8 and 9, respectively.

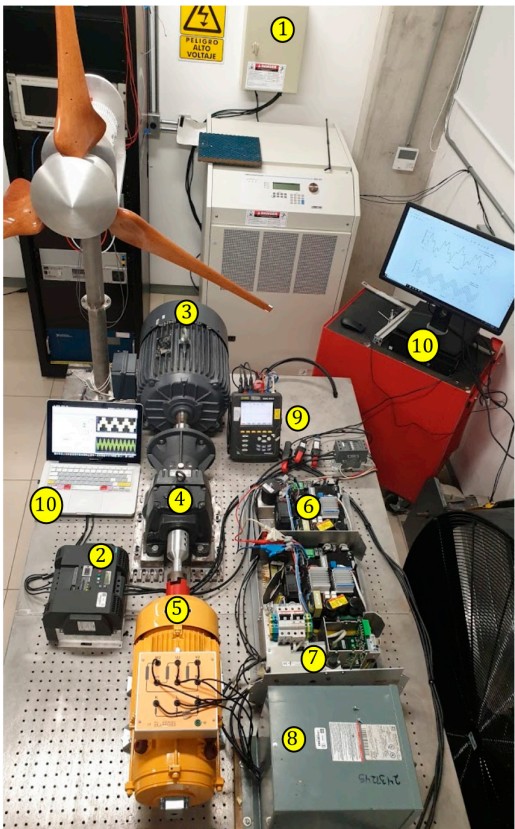

**Figure 8.** Experimental setup of the tested DFIG.

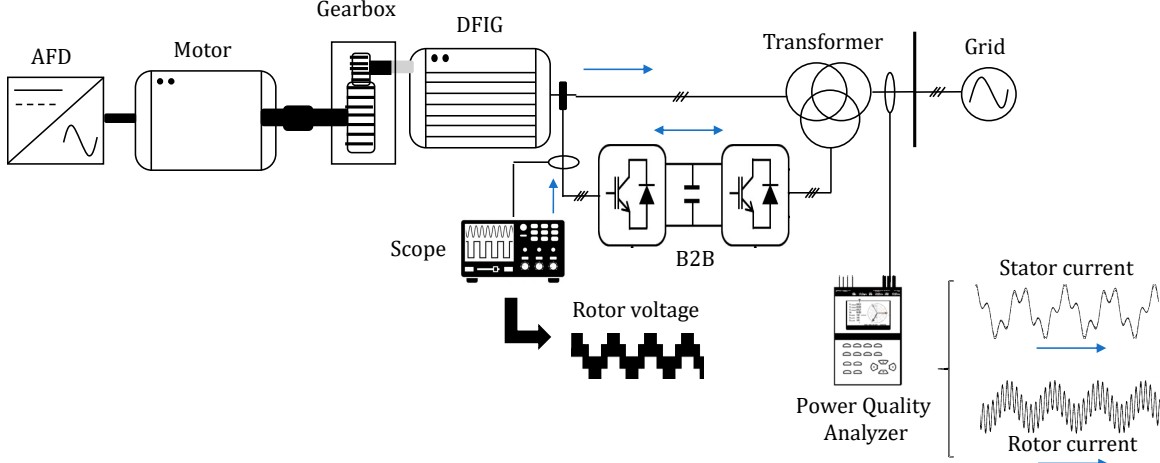

**Figure 9.** Block diagram of the experimental setup.

The detailed parameters of the experimental setup are listed in Table 2. The DFIG parameters are provided in Appendix A.

**Table 2.** Parameters of the experimental setup.

| No. | Equipment | Characteristics |
|---|---|---|
| 1 | Power supply | 450 V, 18 A. |
| 2 | Adjustable Frequency Drive | 480 V, 35.5 A, Motor: 20 HP. |
| 3 | Squirrel cage motor | 10 HP, 460 V, 17.3 A. |
| 4 | Gearbox | Ratio 4.5:1, 12.5 kW |
| 5 | DFIG | 3 kW, 230/400 V Δ/Y, 4.5 A. |
| 6 | Rotor side converter | IGBT Type: 1MBH30D, 4.5 kW, 5 kHz. |
| 7 | Grid side converter | IGBT Type: 1MBH30D, 4.5 kW, 2.5 kHz. |
| 8 | Three-phase transformer | 480 V Y-Y |
| 9 | Power quality analyzer | PowerPad, Mod. 3945-B. |
| 10 | Computer | Software MATLAB/Simulink® |

This system contains an induction motor, a single helical gearbox, an adjustable frequency drive (AFD), a three-phase DFIG wind turbine, and a three-phase B2B power converter. This system involves a 10 HP squirrel-cage induction motor with its rotor coupled directly to a single helical gearbox, where the 3 kW-DFIG is also connected. The speed of the induction motor is controlled by an AFD to simulate different wind speeds.

The rotor winding of the DFIG is connected to the rotor-side converter of the B2B, while the stator winding of the DFIG is connected directly to the electrical grid by means of the three-winding transformer. The grid-side converter controls the DC-link voltage at 160 V. The sampling and switching frequencies for both converters are 5 and 2.5 kHz, respectively.

The results of the model proposed in steady-state are compared with real lab measurements. The machine's steady-state operating point results from the load flow analysis are presented in Table 3, assuming wind speeds *P* and *Q* as input data. Different wind speeds are considered as input data assuming a unit power factor, which means that the grid-side converter transmits active power in both directions (positive and negative), so that the B2B power converter operates with a unit power factor; the reactive power should be zero. If the power factor is positive, the electrical grid consumes the active power from the B2B power converter. The power factor is a unit because the reactive power is zero. In this mode of operation, the current and voltage must be in phase with the electrical grid.



**Table 3.** Steady-state operating condition of the DFIG with P.F. = 1.

| Variables | Wind Speed | | | |
|---|---|---|---|---|
| | 5 **m/s** | 7 **m/s** | 10 **m/s** | 14 **m/s** |
| $P$ | 2070 W | 2250 W | 3450 W | 4140 W |
| $Q$ | 689 Vars | 674 Vars | 1149 Var | 1378 Var |
| $P_s$ | 2484 W | 2577 W | 2906 W | 3116 W |
| $P_r$ | −414 W | −328 W | 564 W | 1048 W |
| $U_s$ | 427 V | 432 V | 436 V | 438 V |
| $\delta_s$ | 88.7° | 126.4° | 36.6° | 274.1° |
| $U_r$ | 212.3 V | 87.1 V | 22.5 V | 146.6 V |
| $\delta_r$ | −63.4° | 25.1° | −123.8 | 40.9° |
| $I_s$ | 1.45 A | 3.833 A | 5.45 A | 10.05 A |
| $\varphi_s$ | 27.6° | −96.1° | 42.2° | −103.2° |
| $I_r$ | 4.35 A | 5.74 A | 9.46 A | 15.07 A |
| $\varphi_r$ | −2.21° | −155° | 49.2° | 127.4° |
| $s$ | −0.188 | −0.06 | −0.756 | −0.16 |

Alternatively, if the power factor is negative, the electrical grid provides active power to the B2B power converter. From the electrical grid, the active power is negative, indicating that active power is generated and consumed by the B2B power converter. Again, in this operating mode, the unit power factor is suitable because the reactive power is zero. Concerning the voltage and current of the electrical grid, a phase shift of 180° exists. Finally, another operation mode of the B2B power converter is analyzed, when the exchange of active power with the electrical grid is minimal such that the B2B power converter is considered to operate with a power factor of zero. In this situation, only the reactive power exchange takes place, without considering the voltage drop in the resistance. Additionally, the current is delayed by 90° relative to the voltage of the electrical grid, indicating the availability of inductive reactive power. The operating condition that enables increasing the available power because of a rise in the wind is considered. The results in Table 3 present an active power, with a minimum increase as the wind speed increases, until the maximum power is reached. Meanwhile, the wind turbine begins to absorb reactive power as the wind speed increases and remains stable when the wind speed attains 10 m/s. Note that the DFIG exhibits a broader range of operation than other wind generation systems because it operates at its nominal speed and at higher or lower synchronization speeds. This is achieved by adjusting its variables to obtain the maximum power before the speed variation.

*6.1. Test Case 1: A 3 kW DFIG-Based Wind Turbine Connected to the Electric Grid*

For the harmonic analysis of the DFIG, the results obtained are directly usable as initial values in a steady-state model of the machine. A wind speed of 10 m/s is considered input data. For this case, the induction generator is excited with a sinusoidal three-phase balanced voltage of 436 V at 60 Hz in the stator winding from the electrical grid and a non-sinusoidal three-phase balanced voltage of 22.5 V at 45 Hz applied to the rotor winding generated by the B2B power converter, which produced the harmonic components of order: 5th, 7th, 11th, 13th, 17th, 19th, 23th, 25th, and 29th, known as the characteristic harmonics of the B2B power converter. The magnitude and angle of the voltage harmonic components are presented in Table 4. The waveform of the rotor voltage is shown in Figure 10; furthermore, current waveforms from measurements and simulations are displayed in Figures 11 and 12, respectively.

**Table 4.** Harmonic components of the rotor voltage for test case 1.

| | Harmonic Components | | | | | | | |
|---|---|---|---|---|---|---|---|---|
| | **5th** | **7th** | **11th** | **13th** | **17th** | **19th** | **23rd** | **25th** |
| Voltage | 4.3 | 3.01 | 1.98 | 1.63 | 1.03 | 0.88 | 0.73 | 0.67 |
| Degrees | −84.3 | 63 | 161.1 | 100.7 | −7.9 | 26.3 | 40.1 | 251.6 |

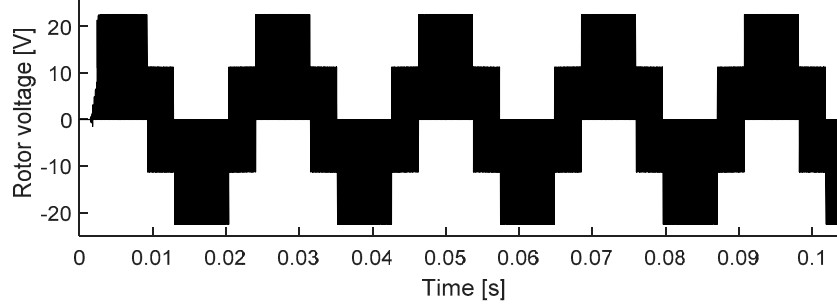

**Figure 10.** Rotor voltage.

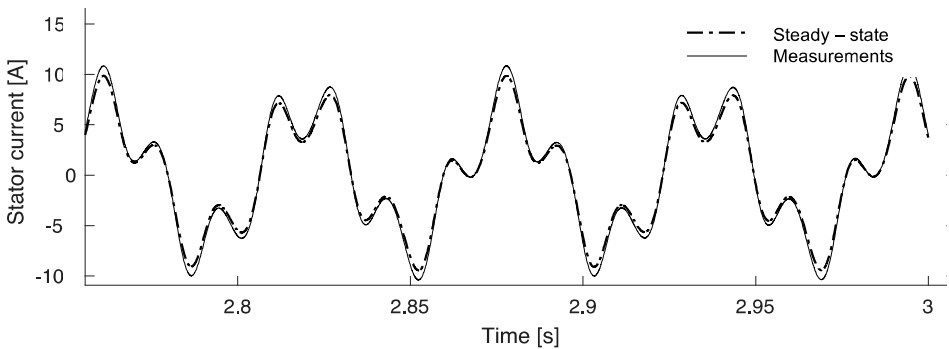

**Figure 11.** Stator current.

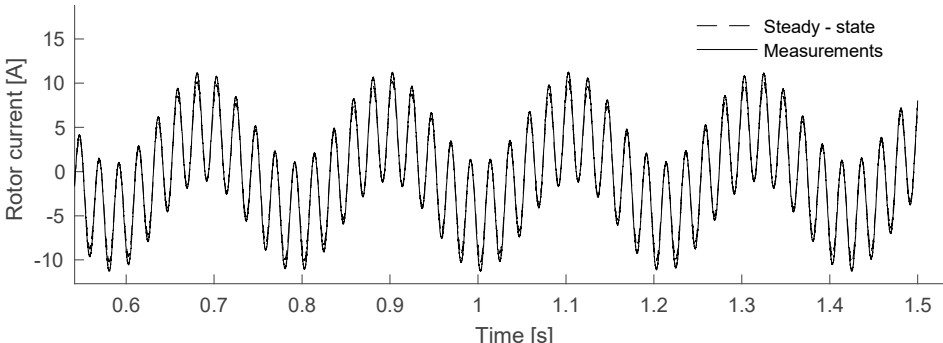

**Figure 12.** Rotor current.

The rotor current harmonic caused by the switching of the rotor-side converter of the B2B power converter appears at frequencies of $f_{rh} = (6h \pm 1)sf_{es}$, $h = 0, 1, 2, 3, \ldots$ i.e., at the 5th, 7th, 11th, 13th ... of the fundamental frequency in the currents of the induction generator as summarized in Table 5, where the current waveforms presented in this study are attained by applying the solution of Equations (38) and (39) in the previous section. Table 5 also shows the current THD index of the current waveforms, both stator and rotor are presented in Figures 11 and 12. The rotor harmonic currents establish rotating magnetic fields in the airgap, inducing currents of corresponding frequencies to the stator winding. The DC current distortion introduced by the grid-side converter is also reflected in the rotor currents, which causes the appearance of additional harmonics. The full harmonic content of the

rotor current is then given by $f_{rh} = |(6h \pm 1)s \pm 6m| f_{es}$, $h, m = 0, 1, 2, 3, \ldots$ setting $m = 0$ yields the "conventional" harmonic frequencies caused by the rotor-side converter. The last expression explains the harmonic frequencies at 45 (fundamental rotor frequency), 225, 315, 495, 585, 765, and 855 Hz, corresponding to $h = 5, 7, 11, 13, 17, 19$.

**Table 5.** Summary of harmonic currents for test case 1.

| STATOR CURRENT | | | | ROTOR CURRENT | | | |
|---|---|---|---|---|---|---|---|
| Sequence | Magnitude | Angle | Frequency | Sequence | Magnitude | Angle | Frequency |
| + | 5.67% | 208.9° | 60 Hz | + | 5.2% | 264.4° | 45 Hz |
| 0 | 12.43% | 191.7° | 180 Hz | − | 4.7% | 2.7° | 225 Hz |
| − | 7.0% | −87.2° | 300 Hz | + | 3.07% | 0.4° | 315 Hz |
| + | 4.91% | 185.2° | 420 Hz | − | 2.31% | 23.1° | 495 Hz |
| − | 3.09% | 53.8° | 495 Hz | + | 1.85% | −84° | 571.2 Hz |
| + | 2.61% | −8.8° | 486.1 Hz | − | 1.55% | 55.2° | 760.3 Hz |
| − | 1.78% | −13.9° | 1040 Hz | + | 1.33% | 149° | 861.2 Hz |
| + | 1.47% | 10.4° | 1166 Hz | − | 1.16% | −0.4° | 988 Hz |
| − | 1.35% | −159° | 1182 Hz | + | 1.03% | 40.3° | 1053 Hz |
| + | 1.17% | 18.7° | 1422 Hz | − | 0.93% | 37.4° | 1126.3 Hz |
| − | 1.09% | 196° | 1529 Hz | + | 0.84% | 325° | 1308.1 Hz |
| + | 0.97% | 270° | 1765 Hz | − | 0.77% | −5.3° | 1401 Hz |
| − | 0.92% | −21.6° | 1901 Hz | + | 0.71% | 250° | 1666 Hz |
| + | 0.79% | −250° | 1966 Hz | − | 0.66% | 73.2° | 1907 Hz |
| − | 0.72% | 333° | 2002 Hz | + | 0.58% | −22.3° | 2002 Hz |
| + | 0.69% | −47.4° | 2245 Hz | | | | |
| **Total Harmonic Distortion (THD) in stator** = 25.9% | | | | **Total Harmonic Distortion (THD) in rotor** = 33.6% | | | |

The same principle is applicable to the harmonics of the grid-side converter, appearing at the frequencies $f_{sh} = |(6m \pm 1)s \pm 6hs| f_{es}$, $h, m = 0, 1, 2, 3, \ldots$, with the "integer" harmonics obtained presented in Table 5. For $h \neq 0$, yields of the sub- and inter-harmonic contents of the recovery cascade current are derived, and these are quite significant depending on the operating slip. The rotor harmonic currents establish rotating magnetic fields in the airgap, inducing currents of corresponding frequencies to the stator winding. Since the slip $s$ is a non-integer, the stator current harmonic content consists mainly of sub- and inter-harmonics, creating undesirable effects on the supply system. For instance, low-frequency sub-harmonics appear as unidirectional components superimposed on the phase currents, while sub- and inter-harmonics near the supply frequency may create a beat effect on the stator current magnitude.

### 6.2. Test Case 2: A 50 kW DFIG-Based Wind Turbine Connected to the Electric Grid

To ensure that the DFIG model for harmonic analysis works with great precision, an additional test case is proposed considering a higher power DFIG, i.e., 50 kW. In this test case, the induction generator is excited with a sinusoidal three-phase balanced voltage of 460 V at 60 Hz in the stator winding coming from the electrical grid and a non-sinusoidal three-phase balanced voltage of 20 V at 45 Hz generated by a B2B power converter. The magnitude and phase angle of the harmonic components of the non-sinusoidal voltage are shown in Table 6. A mechanical torque of 198 N × m was used.

**Table 6.** Harmonic components of the rotor voltage for test case 2.

| | Harmonic Components | | | | | | | |
|---|---|---|---|---|---|---|---|---|
| | **5th** | **7th** | **11th** | **13th** | **17th** | **19th** | **23rd** | **25th** |
| Voltage | 194.7 | 142 | 82.3 | 74.2 | 58.12 | 48.8 | 40.65 | 39.2 |
| Degree | 269.2 | 88.9 | 88.3 | 268 | 267.4 | 87.1 | 86.5 | 266.5 |

It is important to note that only a simulation was performed, but again the results match well between the results obtained from the transient-state model and those obtained from the proposed steady-state model. Equations of the transient-state model are obtained and described in Appendix A. The current waveforms obtained from the simulation are shown in Figures 13 and 14. Table 7 summarizes the harmonic currents in the DFIG for the test case. Table 7 also shows the current THD index of the current waveforms, both stator and rotor presented in Figures 13 and 14. It should be noted that the models were compared after 2 s, which was the time that the model in transient-state reached the steady-state.

**Table 7.** Summary of harmonic currents for test case 2.

| STATOR CURRENT | | | | ROTOR CURRENT | | | |
|---|---|---|---|---|---|---|---|
| **Sequence** | **Magnitude** | **Angle** | **Frequency** | **Sequence** | **Magnitude** | **Angle** | **Frequency** |
| + | 52.1% | −41.2° | 60 Hz | + | 41.43% | 3.6° | 45 Hz |
| − | 12.42% | 90.4° | 313.8 Hz | + | 16.40% | −0.8° | 238.5 Hz |
| + | 7.44% | 68.4° | 463.6 Hz | − | 10.53% | 271° | 325.3 Hz |
| − | 5.42% | −31° | 588 Hz | + | 7.83% | 55.3° | 411.3 Hz |
| + | 4.42% | 77.3° | 683.2 Hz | − | 6.12% | −83° | 515.2 Hz |
| − | 3.98% | 95.6° | 816 Hz | + | 5.11% | 44° | 612 Hz |
| + | 3.40% | −41.6° | 1064.4 Hz | − | 4.41% | 77.5° | 730.3 Hz |
| − | 2.88% | 3.97° | 1158 Hz | + | 3.85% | −3.6° | 823.5 Hz |
| + | 2.46% | −25.3° | 1384.6 Hz | − | 2.79% | 17.8° | 971.8 Hz |
| − | 1.98% | 233° | 1521.2 Hz | + | 2.21% | −99.2° | 1071.63 Hz |
| **Total Harmonic Distortion (THD) in stator** = 17.8% | | | | **Total Harmonic Distortion (THD) in rotor** = 23.1% | | | |

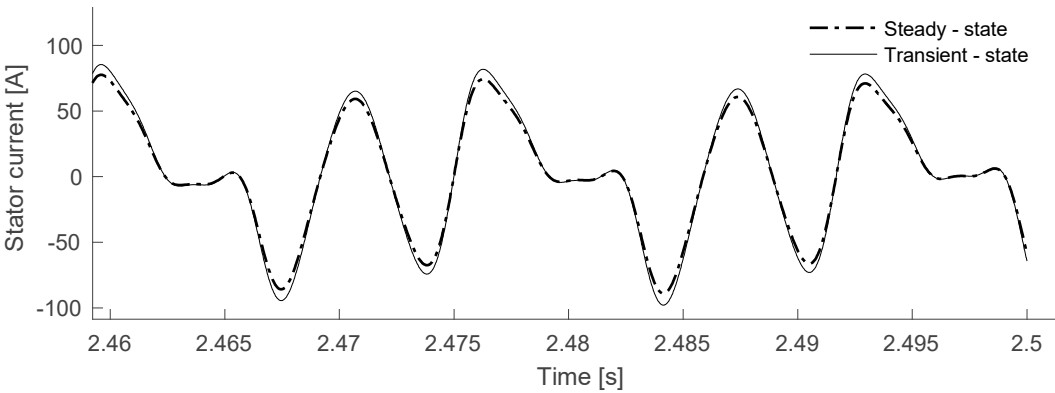

**Figure 13.** Stator current.

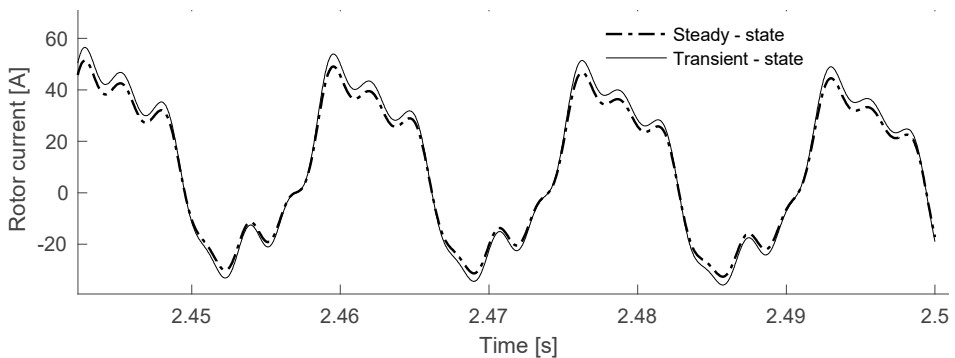

**Figure 14.** Rotor current.

## 7. Conclusions

This paper demonstrates that a grid-connected DFIG model derived by employing the frequency-domain harmonic analysis is able to reproduce the steady-state behavior and the transient dynamics with a high effectiveness since the proposed model suitably matches with the actual measurements of the grid-connected DFIG implemented. The main results are summarized in the following concluding remarks.

1.  The DFIG model at fundamental frequency considering three-phase non-sinusoidal balanced voltage in both windings of the machine to determine the current response was analyzed.
2.  An iterative analysis of the DFIG's steady-state operation was presented. The model is directly applicable to load flow analysis when wind speed, active and reactive powers are considered as input variables. The results are usable as initial values in a steady-state model for the DFIG and internal variables for the generator.
3.  The DFIG model at harmonic frequencies considering three-phase non-sinusoidal voltage in both windings of the machine was analyzed. Harmonic and non- harmonic currents were generated on both sides of the machine depending on the slip and the fundamental frequency.
4.  The power balance in the B2B power converter, as well as the interconnection to the rotor winding of the machine, were analyzed considering that the B2B power converter works in a bidirectional operation.

Finally, the DFIG was connected to the electrical grid for harmonic studies. The results from the proposed model were consistent with those from lab measurements. This paper provides an easy and clear analysis of frequencies generated by the DFIG in the steady-state, yielding a proper model for harmonic and non-harmonic analysis of the DFIG connected to the electrical grid.

**Author Contributions:** Conceptualization, methodology, E.H.-M., and O.A.J.S.; formal analysis, and investigation, E.H.-M.; resources, O.A.J.S.; writing—original draft preparation, E.H.-M., and V.L.; software, and validation, E.H.-M. and R.I.-C. All authors have read and agreed to the published version of the manuscript.

**Funding:** This research received no external funding.

**Conflicts of Interest:** The authors declare no conflict of interest.

## Nomenclature

| | |
|---|---|
| $V_s \angle \delta_s$ and $V'_r/s \angle \delta_r$ | Magnitude and angle of the stator and rotor voltages |
| $I_s \angle \varphi_s$ and $I'_r \angle \varphi_r$ | Magnitude and angle of the stator and rotor currents |
| $r_s$ and $r'_r$ | Stator and rotor resistances |
| $Ll_s$ and $L'l_r$ | Stator and rotor inductances |
| $L_M$ | Magnetizing inductance |
| $\omega_{es}$ | Electric angular velocity, equivalent to $2\pi f_{es}$ |
| $f_{es}$ | Excitation source frequency in the stator |

| | |
|---|---|
| $\omega_r$ | Rotor mechanical velocity |
| $f_{er}$ | Excitation source frequency in the rotor |
| $\theta_{eff}$ | Phase angle between the rotor and stator in steady-state |
| $s$ and $s_r$ | Stator slip and rotor slip |
| $T_e$ | Electric torque, equivalent to $P_e/\omega_r$ |
| $P_m$, $P_s$ and $P_r$ | Mechanical, stator and rotor power, respectively |
| $P_{loss}$ | Losses power, equivalent to $\left(I_s^2 r_s + I_r^2 r_r\right)$ |
| $P$ and $Q$ | Active and reactive powers, respectively |
| $\rho$ | Air density expressed in $[\mathbf{kg/m^3}]$ |
| $A$ | Swept area by the blades expressed in $\left[m^2\right]$ |
| $V_W$ | Wind speed upstream the rotor [m/s] |
| $C_p$ | Power coefficient |
| $\beta$ and $\lambda$ | Pitch angle expressed in [deg] and tip speed ratio |
| $v_t$ | Blade tip speed expressed in [m/s] |
| $f_{sh}$ and $f_{rh}$ | Harmonic frequencies for stator and rotor, respectively |
| $V_{sh} \angle \delta_{sh}$ | Magnitude and angle of the harmonic stator voltage |
| $V'_{rh}/s_h \angle \delta_{rh}$ | Magnitude and angle of the harmonic rotor voltage |
| $I_{sh} \angle \varphi_s$ | Magnitude and angle of the harmonic stator current |
| $I'_{rh} \angle \varphi_{rh}$ | Magnitude and angle of the harmonic rotor current |
| $s_h$ and $s_{rh}$ | Harmonic slip in stator and rotor, respectively |
| $S_1, S_1, \ldots S_6$ | IGBT's switches of the rotor-side converter |
| $S'_1, S'_2, \ldots S'_6$ | IGBT's switches of the grid-side converter |

## Appendix A

The parameters of the three-phase induction generator used for the cases studies are presented below (See Table A1).

**Table A1.** DFIG parameters.

| PARAMETERS | 3 kW | 50 kW |
|:---:|:---:|:---:|
| Poles number | 4 poles | 4 poles |
| Inertia constant $\left(kg/m^2\right)$ | 0.089 | 1.662 |
| Nominal line current | 5.8 A | 46.8 A |
| Nominal line-to-line voltage | 230/400 V | 980 V |
| Nominal torque | $11.9\,N \times m$ | $198\,N \times m$ |
| Nominal frequency | 60 Hz | 60 Hz |
| Stator resistance, $r_s$ | 0.435 Ω | 0.087 Ω |
| Rotor resistance, $r_r$ | 0.816 Ω | 0.228 Ω |
| Stator inductance, $L_{ls}$ | 2 mH | 0.8 mH |
| Rotor inductance, $L_{lr}$ | 2 mH | 0.8 mH |
| Magnetizing inductance, | 69 mH | 34.7 mH |
| Rotor speed | 1500 rpm | 1705 rpm |

The state equations that describe the behavior of symmetrical machine are given by Equation (A1):

$$\begin{bmatrix} p\mathbf{i}_{abcs} \\ p\mathbf{i}'_{abcr} \end{bmatrix} = \begin{bmatrix} \mathbf{L}_s & \mathbf{L}'_{sr} \\ \mathbf{L}'^T_{sr} & \mathbf{L}'_r \end{bmatrix}^{-1} \begin{bmatrix} \mathbf{r}_s & p\mathbf{L}'_{sr} \\ p\mathbf{L}'^T_{sr} & \mathbf{r}'_r \end{bmatrix} \begin{bmatrix} \mathbf{i}_{abcs} \\ \mathbf{i}'_{abcr} \end{bmatrix} + \begin{bmatrix} \mathbf{L}_s & \mathbf{L}'_{sr} \\ \mathbf{L}'^T_{sr} & \mathbf{L}'_r \end{bmatrix}^{-1} \begin{bmatrix} \mathbf{v}_{abcs} \\ \mathbf{v}'_{abcr} \end{bmatrix} \quad (A1)$$

The apostrophe means quantities referring to the stator and $p$ the derivative, and $v_{abcs}$ and $v_{abcr}$ are the phase stator and rotor voltages in volts, respectively. In addition, $i_{abcs}$ and $i_{abcr}$ are the phase stator and rotor

currents in amperes, respectively, while $r_s$ and $r_r$ are the stator and rotor resistances in ohms. $Ll_s$ and $Ll_s$ are the leakage inductances of the stator and rotor in Henries. The general model of the induction machine is given by Equation (A2),

$$\begin{bmatrix} \mathbf{v}_{abcs} \\ \mathbf{v}'_{abcr} \end{bmatrix} = \begin{bmatrix} \mathbf{r}_s & 0 \\ 0 & \mathbf{r}'_r \end{bmatrix} \begin{bmatrix} \mathbf{i}_{abcs} \\ \mathbf{i}'_{abcr} \end{bmatrix} + \rho \begin{bmatrix} \boldsymbol{\lambda}_{abcs} \\ \boldsymbol{\lambda}'_{abcr} \end{bmatrix} \tag{A2}$$

The matrices of stator and rotor resistances are given by Equation (A3)

$$\mathbf{r}_s = \begin{bmatrix} r_{as} & 0 & 0 \\ 0 & r_{bs} & 0 \\ 0 & 0 & r_{cs} \end{bmatrix} \mathbf{r}'_r = \begin{bmatrix} r'_{ar} & 0 & 0 \\ 0 & r'_{br} & 0 \\ 0 & 0 & r'_{cr} \end{bmatrix} \tag{A3}$$

In matrix notation, the flux linkages of the stator and rotor windings $\boldsymbol{\lambda}_s$, $\boldsymbol{\lambda}_r$, in terms of *sr* the winding inductances and currents are given by Equation (A4),

$$\begin{bmatrix} \boldsymbol{\lambda}_{abcs} \\ \boldsymbol{\lambda}'_{abcr} \end{bmatrix} = \begin{bmatrix} \mathbf{L}_s & \mathbf{L}'_{sr} \\ \mathbf{L}'^T_{sr} & \mathbf{L}'_r \end{bmatrix} \begin{bmatrix} \mathbf{i}_{abcs} \\ \mathbf{i}'_{abcr} \end{bmatrix} \tag{A4}$$

By grouping terms, we have Equation (A5)

$$\begin{bmatrix} v_{abcs} \\ v'_{abcr} \end{bmatrix} = \begin{bmatrix} \mathbf{r}_s + p\mathbf{L}_s & p\mathbf{L}'_{sr} \\ p\mathbf{L}'^T_{sr} & \mathbf{r}'_r + p\mathbf{L}'_r \end{bmatrix} \begin{bmatrix} \mathbf{i}_{abcs} \\ \mathbf{i}'_{abcr} \end{bmatrix} + \begin{bmatrix} \mathbf{L}_s & \mathbf{L}'_{sr} \\ \mathbf{L}'^T_{sr} & \mathbf{L}'_r \end{bmatrix} \begin{bmatrix} p\mathbf{i}_{abcs} \\ p\mathbf{i}'_{abcr} \end{bmatrix} \tag{A5}$$

Therefore, Equation (A6) is simplified as,

$$\begin{bmatrix} \mathbf{v}_{abcs} \\ \mathbf{v}'_{abcr} \end{bmatrix} = \begin{bmatrix} \mathbf{r}_s & p\mathbf{L}'_{sr} \\ p\mathbf{L}'^T_{sr} & \mathbf{r}'_r \end{bmatrix} \begin{bmatrix} \mathbf{i}_{abcs} \\ \mathbf{i}'_{abcr} \end{bmatrix} + \begin{bmatrix} \mathbf{L}_s & \mathbf{L}'_{sr} \\ \mathbf{L}'^T_{sr} & \mathbf{L}'_r \end{bmatrix} \begin{bmatrix} p\mathbf{i}_{abcs} \\ p\mathbf{i}'_{abcr} \end{bmatrix} \tag{A6}$$

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
