# Peer review of "Modelling and Validation of a Grid-Connected DFIG by Exploiting the Frequency-Domain Harmonic Analysis"

_applsci, doi:10.3390/app10249014_

Round 1
Reviewer 1 Report
The chosen topic is interesting and the paper is well written and documented, so for my part it is ready to be published
Author Response
Thanks for your comments

Reviewer 2 Report
The article is devoted to a Wind Energy Conversion Systems based on a Doubly–Fed Induction Generator (DFIG). А complete model of the DFIG connected to the electrical grid has been developed. The model is used to study the higher harmonics of current and voltage introduced by DFIG into the electrical grid.
The research has scientific and practical significance. However, it is not clear from the article how the developed model is planned to be used in the future.
Author Response
The model developed in this research presents various capabilities that will be mentioned below:
- This is a clear and concise model of a 3 kW DFIG-based wind turbine connected to the electrical grid for harmonic propagation studies. Additionally, it has the advantage of operating with higher power machines, for example, 5, 10, 50, 500 and up to 2500 kW, obtaining convincing results from harmonic analysis.
- This model can be implemented with a modern control strategy in the back-to-back power converter to mitigate the harmonic components injected into the rotor of the machine.
- The model could contemplate the addition of several wind turbines of different capacities, forming a wind farm connected to the electrical grid for harmonic propagation studies. In addition, different aspects of power quality could be studied and analyzed, such as flickers, voltage imbalances or frequency varia-tion.
- This model could also be easily adapted to obtain a new model of failure analy-sis in DFIG wind turbines connected to the electrical grid, that is, the imple-mentation of effective fault detection systems that minimize their maintenance costs.
- This model could become a useful tool in Universities or in the private sector for the harmonic analysis of wind turbines of different capacities connected to the electricity grid.

Reviewer 3 Report
The paper is well written and presents a useful model of DFIG considering harmonic contributions. The specific comments are as follows:
- Please clearly mention the gap of the existing works.
- How the load flow models would chnage if the presence of single tuned filter is considered to mitigate harmonics?
- What are the advantages/ benefits of the proposed model against the conventional design?
- Please discuss the impact of the proposed model if a utility-scale wind farm is to be connected to a grid?
- Comment on the TDD for the proposed model.
Author Response
Reply to the reviewer
- The analyzed literature does not provide clear and sufficient information about the inter–harmonic content generated in the interconnection of wind farms to the electrical. Some studies have been presented in the literature investigating inter–harmonic effects in DFIG terminal quantities and mechanical signals. For example, the inter–harmonic effects created by higher order stator and rotor supply harmonics in grid connected DFIG systems were investigated using stator and rotor currents, electromagnetic torque and frame acceleration signals measurements. Other papers have examined the general spectral content of various electrical and mechanical signals from DFIGs such as the stator currents, rotor currents, stator active and reactive power. The effects of switching harmo-nics have also been studied using measurements of electromagnetic torque and stator and rotor currents for autonomous DFIG systems. Similarly, studies have been carried out on the spectral content of the voltage and current signals at the generator terminals and at the high voltage points of the interconnections of large capacity (MW) DFIGs, and it was emphasized that the inter–harmonics caused by non-sinuous winding distribution is a major contributor to the inter-harmonic emissions of the DFIG. Therefore, we decided to present a study of the inter–harmonic analysis that occurs in the DFIG when it is excited by a back-to-back power converter in which two influences of the distortion are checked: 1) dependency on the relative power: the lower the relative power, the higher the distortion level, and 2) dynamic of the Wind Energy Conversion: fast load changes lead to jumps in the converter switching frequencies with also increasing distortion.
* NOTE: This information has been inserted into Page 2, column 88 of the document.
- Since the results of the THD index of the generator currents of this investigation exceed the limit established by the grid codes, it is necessary to implement a harmonic compensation method, for example, a single tuned filter that reduces, considerably, harmonic distortion and correction for low power factor. For this, the electrical characteristics of the grid must be considered, as well as the reso-nance points present in it, in order to avoid possible damage to the single tuned filter components and to insert extra resonance levels not desired on the grid. For this purpose, the design parameters of the single tuned filter must be consi-dered and related to the parameters of the generator and the grid components, which would modify the mathematical modeling of the load flow models. The variables that the filter design parameters must consider are the active (10) and reactive powers (11), respectively.
* NOTE: This information is not considered in the body of the document, it only constitutes a response to the reviewer.
- The advantages and benefits of the model proposed in this research with respect to the conventional models existing in the literature are listed below:
- Our model proposes a harmonic analysis of a 3 kW DFIG–based wind turbine connected to the electrical grid caused by nonsinusoidal conditions in the rotor. The components generated in the machine suggest the presen-ce of inter–harmonic components which arise from the interaction of the stator and rotor windings of the machine.
- This model has the ability to analyze and calculate the harmonic compo-nents, from which the torques produced by these interactions between stator and rotor harmonic components can be found. During unbalanced stator conditions, symmetric component theory is applied to the stator voltage to get positive, negative, and zero sequence components of stator and rotor currents. In both scenarios, the harmonic components of the electromagnetic torque can be calculated from the interactions of the har-monic components of the stator and rotor currents.
- This model has the ability to operate with DFIG–based wind turbines of different powers, that is, from low powers to MW capacities. Therefore, it turns out to be a flexible and useful model for different teaching methods in Universities.
* NOTE: This information is not considered in the body of the document, it only constitutes a response to the reviewer. Unless the reviewer deems it to be added to the main document.
- The model proposed in this research offers the alternative to considering one or more wind turbines of different capacities for connection to the electrical grid. The analysis of the behavior of the electrical grid when one or various wind turbines are connected must consider possible disturbances to the existing elec-trical system or to the installation itself, which would motivate further main-tenance of the unit. It should be noted that this document highlights the impor-tance of the small wind power industry, so the experimental validation is carried out with a 3 kW DFIG–based wind turbine. However, this model could be extra-polated to higher capacity wind generators (50, 500 and 2500 HP). An important observation to note is that the harmonic distortion in rotor voltage, stator and rotor currents, frequency, active, reactive, and apparent power, and power factor of plants wind power at the common coupling point (PCC) with the high-voltage transmission grid decrease considerably. This is because, as the number of wind turbines and their power increases, the system becomes more robust and the harmonic injection from the back-to-back power converter is attenuated. Therefore, if a large capacity wind farm is considered, the harmonic impact will be less at the PCC.
* NOTE: This information is not considered in the body of the document, it only constitutes a response to the reviewer. Unless the reviewer deems it to be added to the main document.
- Harmonic indices have been developed to assess the service quality of a power system with respect to the harmonic distortion levels. These indices are measu-res of the effective value of a waveform and can be applied to both the current and the voltage. The IEEE–519 document has set limits on the level of allowable harmonics (IEEE Standard 519-1992, 1992). Several indices are available for harmonic analysis; however, the two most commonly used in wind turbines are the total harmonic distortion (THD) and the total demand distortion (TDD). The latter is defined as the total harmonic current distortion defined by the ratio of the RMS value of the sum of the individual harmonic amplitudes to the maxi-mum or rated demand load current. So, knowing that TDD is the THD of the current (using an average measurement period of 15 or 30 minutes) normalized to the maximum demand load current, then the TDD index is equal to only under load conditions complete. Therefore, the TDD indices of the stator and rotor currents in this research are obtained from the THD indices of the stator and rotor currents considering the full load conditions. It should be noted that the TDD of the current injected from the wind turbine as well as the voltage THD at the connection point of the wind turbine depends on the wind speed, that is, if it is constant or variable. This is important to define if the values of the injected current TDD and the terminal voltage THD are within the allowable range.
* NOTE: This information is not considered in the body of the document, it only constitutes a response to the reviewer.
